# Individual differences in thinking style and dealing with contradiction: The mediating role of mixed emotions

**David Santos**[1]*, **Blanca Requero**[2], **Manuel Martín-Fernández**[3]

**1** IE School of Human Sciences and Technology, IE University, Madrid, Spain, **2** Psychology Department, Universidad Villanueva, Madrid, Spain, **3** Psychology Department, Universidad Autónoma de Madrid, Madrid, Spain

* david.santos@ie.edu

**Data Availability Statement:** All data and code can be found at https://osf.io/yadft/?view_only=f1f8cefed3f144a9bd7f8d3a888dc476.

**Funding:** The author(s) received no specific funding for this work.

## Abstract

The present research examined how individuals' thinking style (holistic vs. analytic) is associated with the way they deal with contradictory information and whether experiencing mixed emotions can mediate this relationship. Participants first completed the thinking style measure and then were exposed to two contradictory pieces of information (Studies 1 and 2). In study 2, we also measured the experience of mixed emotions to test the mediating role of this variable. Across two studies, we found that individuals with a holistic thinking style were more able to reconcile contradictory information compared to individuals with an analytic thinking style. Study 2 showed that the relationship between thinking style and dealing with contradiction was mediated by the experience of mixed emotions. This research extends previous findings on confrontation of contradiction and mixed emotions by using an individual-differences rather than a cultural-differences approach, and establishes mixed emotions as a plausible mediating variable.

## Introduction

Although people deal with contradictions on almost a daily basis, the tools used to resolve these inconsistencies in our social worlds are often very different. Understanding contradiction is important because it may affect people's well-being and physical health [1,2]. Within the field of psychology, cognitive conflict is a well-documented construct that has given rise to many diverse theories in order to account for how individuals respond to its effects [3–7]. The present study examines the relationship between thinking style and the way people deal with contradiction and proposes a new explanatory mechanism, namely the activation of mixed emotions.

Contemporary theories of thinking style address different ways of dealing with contradiction depending on whether the mode of thinking is holistic or analytic [8–10]. These theories define holistic-analytic thinking style as a multifaceted construct composed by several dimensions. The first dimension is causality, which considers the presence of complex causalities and the elements of the universe as interconnected and interrelated (vs. the universe consists of elements that are independent of each other, [11]). The second one is attitudes towards

**Competing interests:** The authors have declared that no competing interests exist.

contradiction, which assesses the preference for resolving the contradiction through a reconciliation strategy, seeking the "middle way" between opposing propositions (vs. contradictions are resolved by choosing one of the two opposite propositions, [12]). The third, perception of change, refers to the tendency to perceive the elements as being in constant change and unpredictable (vs. linear changes and predictable, [13]). The last dimension is locus of attention, which places the focus on "the big picture", considering the elements of the stimulus as a whole (rather than decomposing the stimulus in their parts, ignoring the context, [14]).

Much of the literature on thinking style has focused on differences between cultures. Scholars in many disciplines have demonstrated that people from East Asian cultures tend to have a relatively holistic cognitive orientation, emphasizing relationships and connectedness, perceiving phenomena in constant change, and tending to focus on the context. Conversely, people from Western cultures usually hold a relatively analytic world view, tending to consider that elements' properties remain stable and differentiated, maintaining a linear view about the world, and paying more attention mainly to the object than to the field to which it belongs [8,9,15–17].

Regarding cognitive conflict, a holistic thinking style is characterized by the recognition of contradiction and for being able to maintain multiple perspectives at the same time. That is, this style proposes the search for a "middle ground" between opposing propositions (i.e., *principle of contradiction*: the idea that even two opposite propositions may be plausible [9]). In contrast, analytic thinking is guided by the law of non-contradiction, whereby it is not possible to accept two contradicting propositions at the same time [18,19]. Note that we refer to contradiction as occurring when two pieces of information are inconsistent with each other in such a way that if one of them is true, then it is likely that the other is false [20,21]. When confronted with an apparent contradiction, individuals with a holistic approach will often attempt to reconcile the opposing propositions. In contrast, individuals with an analytic approach will often seek to determine which of the propositions is more plausible and which is less plausible.

As an illustration, Peng and Nisbett [9] presented participants with brief descriptions about contradictory research findings on various topics. Although the opposing statements were superficially incompatible, they did not involve contradictions that were impossible to reconcile. For example, one statement maintained that the diets of people who live a long life consisted of certain types of white meat, whereas the other statement maintained that it was much healthier to be a strict vegetarian. After reading each pair of statements, participants reported how much they believed each of the statements to be true. Results showed that holistic thinkers (i.e., Chinese) rated both statements as more plausible compared to analytic thinkers (i.e., Americans), therefore considering both statements might be true. That is, they used a compromise strategy to deal with the contradiction. In contrast, analytic thinkers used a differentiation strategy that led them to perceive and report one statement as more plausible than the other. The purpose of the present work is to extend these results regarding the reconciliation of opposing information by applying an individual-differences perspective rather than relying on a cross-cultural approach. Thus, when presented with opposing information, holistic thinkers will use a compromise strategy, finding both sides to be plausible, whereas analytic thinkers will use a differentiation strategy, deciding which of the sides is correct. Furthermore, this work aims to incorporate a new explanatory mechanism for this phenomenon, which is the experience of mixed emotions.

An additional characteristic of the holistic thinking style also related to the reconciliation of contradiction is that holistic thinkers are more willing to tolerate "mixed" or "dialectical" emotions compared to analytic thinkers [22,23]. Specifically, previous research has shown that holistic thinkers generally experience more contradictory emotions [24–26], hold more favorable attitudes toward them, and are more comfortable with the simultaneous activation of affective opposites [27,28]. This led holistic thinkers to show null, weak, or even positive

correlations between positive and negative emotions [22,23,26,29]. By comparison, analytic thinkers show a negative correlation between the simultaneous activation of positive and negative emotions, such that individuals who report experiencing pleasant feelings do not report simultaneously experiencing unpleasant feelings (or vice versa).

As people with more holistic thinking are more comfortable by tolerating contradictory or "mixed" emotions compared to those with more analytic style, one might wonder if experiencing mixed emotions can be an element that influences the type of strategy to deal with the contradiction. That is, as holistic thinking is associated with a balance of opposing feelings, this could facilitate individuals to process contradictory information about any object or event using a compromise strategy. In contrast, those with more analytic thinking would use a differentiation strategy rejecting the perspective that makes them feel most uncomfortable. Indeed, previous research has suggested that this idea linking contradiction with mixed emotions might be plausible theoretically, but no prior empirical evidence has been shown to test it. For example, Miyamoto and colleagues [30] suggest that dialecticism in thinking, which emphasizes contradiction, might be in turn reflecting dialecticism in feeling. Thus, this work examines the role of mixed emotions as a mediator between the thinking style and the way people deal with contradiction. Specifically, we propose that the ability to simultaneously experience mixed emotions can be a mechanism that favors the reconciliation of contradiction. As previously noted, individuals with a holistic thinking style (vs. analytic thinking style) can better tolerate the experience of opposite or "mixed" emotions, which may leave them better equipped to reconcile cognitively conflicting information.

Contemporary research has predominantly examined thinking style (holistic and analytic) using a cross-cultural approach, often comparing Western cultures (North-American individuals) with Eastern cultures (e.g., Korean or Chinese individuals, see [31,32]). However, cultural theorists have also suggested that analytic-holistic thinking can be viewed as an individual-difference variable; that is, individuals within a given culture might be more holistic or analytic thinkers [33]. In an effort to test this assumption, Uskul et al. [34] compared farmers, fishers, or herders as a function of their attention styles (analytic vs. holistic). Results suggested that farmers and fishers tended to have more holistic attention style as compared to their herder counterparts. Thus, in the same way as thinking style can vary at a broad level across cultures, so too can thinking style vary on an individual level within the same culture. In a recent application of the Analysis-Holism Scale (AHS [33]), Zhou and colleagues [35] showed that participants with a more holistic thinking style donated more to causes like Covid-19.

Based on these findings, we expect that within the same culture, the holistic-analytic continuum of thinking style [33] will be able to distinguish between those individuals who reconcile contradictory information (i.e., more plausibility) *versus* those who do not. Furthermore, we expect that holistic thinkers are more likely to show mixed emotions, whereas analytic thinkers are expected to show more emotions of the same valence (pleasant or unpleasant). The experience of mixed emotions is proposed as a mediator of the relationship between thinking style and the strategy to confront contradiction. That is, those who simultaneously experience mixed emotions will be better able to adopt a compromise approach toward contradictory information. Therefore, our research extends prior literature in two important ways. First, it applies an individual-differences perspective instead of a cross-cultural approach. Second, it establishes the mediational role of mixed emotions in the relationship between thinking style and the way people deal with contradiction.

## Overview of the present research

The first goal of this research was to explore how individuals' thinking style (holistic vs. analytic) is associated with the way of confronting contradictory information, using an individual-

differences perspective. In particular, we predicted that individuals with a holistic thinking style would use a compromise approach to deal with contradictory information compared to individuals with an analytic thinking style. In Study 1, participants first completed the locus of attention subscale from the AHS [33]. Next, they were exposed to two apparently contradictory pieces of information. The extent to which they considered both pieces of information as plausible was taken as evidence of using a compromise strategy. The second goal of this work was to test whether the experience of mixed emotions could account for the predicted individual difference in the way people deal with contradictory information. Therefore, Study 2 sought to test the mediating role of mixed emotions. We predicted that holistic thinkers would experience more mixed emotions than analytic thinkers, and that this would mediate the relationship between thinking style and the confrontation of contradiction.

## Study 1

The goal of Study 1 was to examine how individuals' thinking style (holistic vs. analytic) is associated with the way they deal with contradictory information. Specifically, we predicted that individuals with a holistic thinking style would tend to adopt a compromise approach toward contradictory information compared to individuals with an analytic thinking style.

### Method

**Participants and design.**   Sixty hundred eighty-five (685) individuals from the United States participated voluntarily (48.9% females, $M_{age}$ = 33.42, $SD$ = 11.60). There were no missing data because we used the force option in the Qualtrics software. We conducted a structural equation model to assess whether holistic-analytic thinking style could predict participants' ability to deal with two contradictory statements (i.e., plausibility in contradictions).

**Procedure.**   Participants were recruited from Amazon Mechanical Turk (MTurk, a crowdsourcing website) in exchange for monetary compensation. Participants completed the study on Qualtrics. First, they completed the locus of attention subscale from the AHS, then read two statements, each of which contained contradictory information. Next, participants were instructed to indicate how much they believed each of the statements to be true. Finally, participants answered several demographic questions, then were debriefed about the purpose of the study. The institutional review board approved this study (IE Research Committee, number IERC/39-2019-2020). Written informed consent was obtained from participant.

**Predictor variable.**   *Thinking style*. Participants thinking style was measured using the 6-item locus of attention subscale from the AHS [33]. We chose this subscale because it captures the differential ability of thinking to integrate incongruent characteristics of an object into a cohesive whole. Beyond the convenience of having a shorter measure of individual differences in thinking style [36,37], previous research has relied on this particular subscale as the dominant measure for evaluating cognitive style-relevant outcomes [38–42]. Participants indicated how much they agreed with items such as "The whole, rather than its parts, should be considered in order to understand a phenomenon," and "It is more important to pay attention to the whole than its parts" on a 7-point Likert-type scale anchored at 1 ("*Strongly disagree*") to 7 ("*Strongly agree*"). The internal consistency in this sample was appropriate ($\alpha$ = .83). Higher scores indicate greater holistic thinking style and lower scores indicate greater analytic thinking style. Values ranged from 1 to 7.00 ($M$ = 5.29, $SD$ = 1.37).

**Outcome variable.**   *Plausibility in contradictions*. Information in each of the two statements were presented as brief descriptions of the findings from a scientific study (adapted from [9], Study 5). The two opposing statements were apparently incompatible. Specifically, the statements were "A health magazine survey found that people who live a long life eat some

sorts of white meat, e.g., fish or chicken" and "A study by a health organization suggests that it is much more healthy to be a strict vegetarian who does not eat meat at all." All participants were instructed to indicate how much they believed each of the statements to be true on two 7-point scales from 1 ("*Strongly disbelieve*") to 7 ("*Strongly believe*").

A 2 × 2 matrix was created with the responses to these two statements, and we coded the responses to reflect plausibility in contradictory statements (coded as 1) and non-plausibility in contradictory statements (coded as 0). On the one hand, participants who responded above the mid-point (>4) to both statements were coded as 1 (n = 391), indicating that they agreed with both statements and found them equally plausible. On the other hand, participants who responded above the mid-point (>4) to one of the statements and below the mid-point (<4) to the other were coded as 0 (n = 193), indicating that they agreed with one statement and disagreed with the other so they found them incompatible. Responses in the mid-point (4) or below (<4) to both statements were dropped from the study, as they can reflect indifference or doubt toward the content of the statements (n = 101). The final sample of this study was hence comprised by 584 participants (42.6% females, $M_{age}$ = 33.18, $SD$ = 11.71).

**Statistical analysis.** First, a descriptive analysis of the 6-item locus of attention subscale of the AHS and the measure of plausibility in contradiction and its indicators (i.e., the statements) was conducted. The means, standard deviations, skew and kurtosis statistics were computed, as well as the correlation between both statements.

To examine how thinking style (holistic vs. analytic) is associated with how individuals deal with contradictory information, a structural equation modeling (SEM) was used. SEM allows researchers to control the measurement error, estimate a latent variable and test whether a specific structural relation between variables is supported by the data [43]. In a SEM analysis, the first step is to estimate a model evaluating the goodness of fit of the latent variable (i.e., measurement model). The second step is to test the structural relations between two or more variables (i.e., structural model). In order to test the measurement model of the thinking style items, a confirmatory factor analysis (CFA) was first conducted using weighted least squares with means- and variances-adjusted (WLSMV) in the sample. A structural model was carried out afterwards to assess whether the thinking style could predict plausibility in contradictions. Given the dichotomous nature of the outcome variable (i.e., plausibility in contradictions), we used a probit regression, where we model participants' probability of agreement with both contradictory statements. Comparative Fit Index (CFI), Tucker–Lewis Index (TLI), and Root Mean Square Error of Approximation (RMSEA) were considered to assess the goodness-of-fit of the models according to the cut-off points established in the literature [43,44]: CFI ≥ .95, TLI ≥ .95, and RMSEA ≤ .06. All analyses were conducted using the free statistical software R [45] and the lavaan library [46]. All data and code can be found at https://osf.io/yadft/?view_only=f1f8cefed3f144a9bd7f8d3a888dc476.

## Results

**Descriptive analysis.** First, a descriptive analysis of the variables of the present study was carried out (see Table 1). The descriptive statistics from the locus of attentions subscale showed that, on average, participants tend to agree with the items, although the standard deviation and low skew and kurtosis statistics (below |-2|) suggested that there was enough variability across the agreement and disagreement categories of the items. Regarding the statements, the descriptive statistics pointed out that participants tend to slightly agree with the statement A more than with the Statement B, although there also was some variability in the responses. The correlation between both statements was weak and negative ($r$ = -.17, $p$ < .001), indicating that, on average, participants who agree with the first statement, tend to disagree with the

**Table 1. Descriptive statistics of the variables from Study 1.**

|  | M | SD | Min | Max | Skew (s.e.) | Kurtosis (s.e.) |
|---|---|---|---|---|---|---|
| i1 | 5.24 | 1.31 | 1 | 7 | -0.92 (.05) | 0.94 (.05) |
| i2 | 5.19 | 1.47 | 1 | 7 | -0.72 (.06) | 0.02 (.06) |
| i3 | 5.17 | 1.43 | 1 | 7 | -0.79 (.05) | 0.37 (.05) |
| i4 | 5.07 | 1.41 | 1 | 7 | -0.66 (.05) | 0.08 (.05) |
| i5 | 5.27 | 1.35 | 1 | 7 | -0.88 (.05) | 0.76 (.05) |
| i6 | 5.77 | 1.08 | 1 | 7 | -0.86 (.04) | 0.85 (.04) |
| StatementA | 5.14 | 1.48 | 1 | 7 | -0.94 (.06) | 0.43 (.06) |
| StatementB | 4.86 | 1.70 | 1 | 7 | -0.65 (.06) | -0.39 (.06) |
| Plausibility in Contradiction | 0.62 | 0.49 | 0 | 1 | - | - |

*Note*: i = item from the locus of attention subscale of the AHS; M = Mean;SD = standard deviation; Min = Minimum; Max = Maximum; s.e. = standard error from the Skew and Kurtosis statistics.

second. The plausibility in contradiction index showed that 62% of the participants were categorized as individuals considering both statements as plausible, whereas the 38% left was categorized as individuals considering one of the statements more plausible than the other.

**Structural equation models.** We conducted a CFA to test the latent structure of the thinking style items, estimating a one-factor model (see Table 2). The goodness of fit of the proposed model was fair, although some fit indices were below the cut offs: $\chi^2$ (9) = 32.07, CFI = .95, TLI = .92, RMSEA (95% CI) = .071 (.045, .098). Therefore, we recalibrated the model by using modification indices [47–49]. According to modification indices, we allowed the errors of two items to correlate (i5: "It is not possible to understand the parts without considering the whole picture", and i6: "We should consider the situation a person is faced with, as well as his/her personality, in order to understand one's behavior"). The modified CFA model displayed an adequate fit: $\chi^2$ (8) = 17.61, CFI = .98, TLI = .96, RMSEA (95% CI) = .048 (.017, .079) (we will use this modification hereafter). Item factor loadings were all equal or above .30 (see Fig 1A for factor loadings).

Next, we estimated a structural model with plausibility in contradiction as the outcome variable and thinking style as the predictor variable (see Fig 1B). The proposed model obtained adequate fit: $\chi^2$ (13) = 41.98, CFI = .96, TLI = .94, RMSEA (95% CI) = .066 (.044, .089). The path from thinking style to plausibility in contradiction was significant, $B$ = 0.39, $SE$ = 0.05, $p$ < .001, indicating that holistic individuals were more likely to adopt a compromise strategy than analytic individuals. In particular, for participants two standard deviations above the mean in the thinking style continuum (i.e., more holistic thinking style), the probability of finding plausibility in both statements was 0.784, whereas for those participants one standard deviation above the mean, the expected probability was 0.653. In contrast, for participants two

**Table 2. Comparison of goodness-of-fit indices across models.**

| Model | $\chi^2$ | df | CFI | TLI | RMSEA (95% CI) | SRMR |
|---|---|---|---|---|---|---|
| Measurement model | 32.07 | 9 | 0.95 | 0.92 | 0.071 (0.045, 0.098) | .053 |
| Measurement model (i5 ~~ i6) | 17.61 | 8 | 0.98 | 0.96 | 0.048 (0.017, 079) | .028 |
| Structural model TS | 41.98 | 13 | 0.96 | 0.94 | 0.066 (0.044, 089) | .035 |

*Note*: TS model: Latent model for Thinking Style; Structural model: SEM model for Thinking Style on Plausibility in Contradiction; ~~: Error term correlation between items $i_5$ and $i_6$; χ2 = Chi-square; df = degrees of freedom; CFI = Comparative Fit Index; TLI = Tucker-Lewis Index; RMSEA = Root Mean Square Error of Approximation; SRMR = Standardized Root Mean Residual.

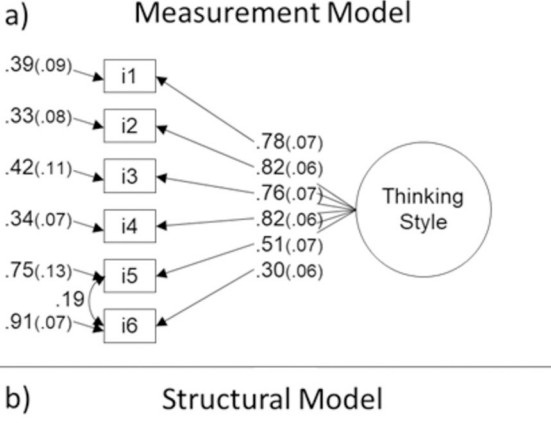

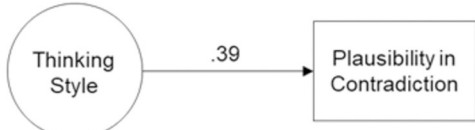

**Fig 1.** (a) top panel: Item factor loadings, Study 1; (b) bottom panel: Structural model, Study 1.

standard deviations below the mean in thinking style (i.e., more analytic thinking style), the probability of finding plausibility in both statements was 0.216, and for those one standard deviation below the mean, this probability was 0.347.

Another procedure to compute the way people deal with contradictions is to obtain the sum of each of the scores for both statements to create one overall index of plausibility in contradictory information, such that higher scores indicated that the two statements were perceived as highly plausible and lower scores indicated that at least one of the statements was perceived as implausible. When we used this other index, the results were similar and the final structural model fitted: $\chi^2$ (13) = 35.81, CFI = .97, TLI = .95, RMSEA (95% CI) = .059 (.041, .077).

## Discussion

This study showed that the structural model obtained appropriate goodness of fit indices. These results supported our hypothesis that not only cultural differences, but also individual differences in thinking style are associated with plausibility in contradictions [9]. Specifically, participants with a holistic thinking style were more likely to find plausibility in contradictions than participants with an analytic thinking style. However, Study 1 does not provide evidence of the process driving this effect. Thus, Study 2 was conducted to replicate the pattern found in Study 1 as well as provide mediational evidence for this phenomenon.

## Study 2

The goal of Study 2 was to explore the mediating role of mixed emotions in the relationship between thinking style and the way people deal with contradiction. We expected to replicate the finding that individuals with a holistic thinking style tend to compromise in dealing with contradictory information compared to individuals with an analytic thinking style. Moreover, we predicted that this relationship would be mediated by mixed emotions, such that holistic (vs. analytic) individuals would report experiencing more mixed emotions.

# Method

**Participants and design.**   Sixty hundred twenty-four (624) individuals from the United States participated voluntarily (50% females, $M_{age}$ = 34.50, SD = 11.44) via Mturk. There were no missing data because we used the force option in the Qualtrics software. We used a SEM including a mediation model to examine the effect of thinking style on dealing with contradictory information through the experience of mixed emotions as a mediator.

**Procedure.**   The procedure was similar to Study 1. Participants were recruited from Amazon Mechanical Turk in exchange for monetary compensation. Participants completed the study on Qualtrics. First, they completed the locus of attention subscale from the AHS [33], then were asked to complete a measure of mixed emotions experienced. Next, participants read two statements with contradictory information. After that, participants were instructed to indicate how much they believed each of the statements to be true. Finally, participants answered several demographic questions, then were debriefed about the purpose of the study. The institutional review board approved this study (IE Research Committee, number IERC/39-2019-2020). Written informed consent was obtained from participant.

**Predictor variable.**   *Thinking style*. Participants' thinking style was measured using the same scale as in Study 1 [33]. Values ranged from 1.83 to 7.00 (M = 5.19, SD = 1.42). The internal consistency in this sample was appropriate ($\alpha$ = .84).

**Mediating variable.**   *Mixed emotions*. To assess mixed emotions experienced, participants were asked to rate how much they felt each of six emotions by using a scale ranging from 1 ("*Not at all*") to 7 ("*Extremely*"). Specifically, we included three pairs of emotions, opposite each other in valence and activation, namely, *contented-upset*, *calm-tense*, *relaxed-nervous*. This measure was adapted from the classic circumplex model of emotion [50–53]. Positive affect (PA) and negative affect (NA) were separately aggregated by averaging the three corresponding items. The internal consistency of both measures was fair ($\alpha$ = .90 for PA; $\alpha$ = .76 for NA). To calculate scores for mixed emotions experienced we used the absolute difference between positive and negative scores, |PA − NA| [54,55]. This measure ranged from 0 to 6, with higher scores on this variable indicating fewer mixed emotions experienced and values closer to 0 indicating more mixed emotions experienced.

**Outcome variable.**   *Plausibility in contradictions*. As in Study 1, participants were exposed to the same contradictory brief descriptions. Again, all participants reported how much they believed each of the statements to be true on two 7-point scales from 1 ("*Strongly disbelieve*") to 7 ("*Strongly believe*").

A 2 × 2 matrix with the responses to these two statements was also computed for this study, and we coded the responses to reflect plausibility in contradictory statements (coded as 1) and non-plausibility in contradictory statements (coded as 0). Participants who responded above the mid-point (>4) to both statements were coded as 1 (n = 234), and participants who responded above the mid-point (>4) to one of the statements and below the mid-point (<4) to the other were coded as 0 (n = 187). Responses in the mid-point (4) or below (<4) to both statements were also dropped from the study (n = 203). The final sample of this study was composed for 421 participants (50.5% females, $M_{age}$ = 34.53, SD = 11.92).

**Statistical analysis.**   As in Study 1, we first carried out a descriptive analysis of the variables involved in this second study. We used, afterwards, a SEM approach and conducted a series of CFAs to assess the measurement model and the structural model. In the measurement model, we tested the latent structure of the thinking style items. In the structural model, we examined the effect of holistic-analytic thinking style on the way people deal with contradictory information through the experience of mixed emotions as a mediator. All models were estimated with WLSMV. We used the same combination of fit indices as in study 1 to assess

the goodness of fit of the models (CFI/TLI ≥ .95, RMSEA ≤ 0.06 [43,44]). All analyses were conducted with the free statistical software software R [45] and the lavaan library [46].

## Results

**Descriptive analysis.** We first conducted a descriptive analysis of the main variables of the second study (see Table 3). All items from the locus of attention subscale presented means around 5, with standard deviations around 1.30, indicating that respondents tended to agree with the items, although the variability was wide. The skew and kurtosis statistics also suggested this trend, with relative plain curves centered around 5 (i.e., where the "somewhat agree" category begins).

The responses to the negative affect items (i.e., "I feel nervous/tense/upset") were below 3, with standard deviations around 2, whereas the responses to the positive affect items (i.e., "I feel calm/contented/relaxed") were around 5, with standard deviations around 1.70. This suggest that, on average, the respondents tended to agree more with the positive affect items than with the negative affect items.

Regarding the contradictory statements, in this study the participants tended to select, on average, the intermediate categories. The correlation between the statements was again negative ($r$ = -.24, $p$ < .001). The plausibility in contradiction index showed that 56% of the respondents was categorized as individuals considering both statements as plausible, whereas the 44% left was categorized as individuals considering one of the statements more plausible than the other.

**Structural equation models.** We tested first the measurement model, estimating one factor for the items of thinking style. As shown in Table 4, the same modified CFA model as in Study 1 for thinking style displayed an adequate fit, $\chi^2$ (8) = 12.12, CFI = .99, TLI = .99, RMSEA (95%) = 0.029 (.001, 0.060). Item factor loadings were all above .30 for thinking style (see Fig 2A for factor loadings).

To test the mediating role of mixed emotions on the relationship between thinking style and the plausibility in contradiction, we estimated a structural model with plausibility in

**Table 3. Descriptive statistics of the variables from Study 1.**

| | M | SD | Min | Max | Skew (s.e.) | Kurtosis (s.e.) |
|---|---|---|---|---|---|---|
| i1 | 5.23 | 1.29 | 1.00 | 7.00 | -0.82 (0.05) | 0.67 (0.05) |
| i2 | 5.05 | 1.51 | 1.00 | 7.00 | -0.50 (0.06) | -0.37 (0.06) |
| i3 | 5.07 | 1.55 | 1.00 | 7.00 | -0.82 (0.06) | 0.23 (0.06) |
| i4 | 4.97 | 1.54 | 1.00 | 7.00 | -0.52 (0.06) | -0.37 (0.05) |
| i5 | 5.15 | 1.39 | 1.00 | 7.00 | -0.70 (0.06) | 0.23 (0.06) |
| i6 | 5.67 | 1.11 | 1.00 | 7.00 | -0.70 (0.04) | 0.55 (0.04) |
| Nervous | 2.67 | 1.99 | 1.00 | 7.00 | 0.78 (0.08) | -0.90 (0.08) |
| Tense | 2.80 | 2.04 | 1.00 | 7.00 | 0.77 (0.08) | -0.82 (0.08) |
| Upset | 2.90 | 2.14 | 1.00 | 7.00 | 0.69 (0.09) | -1.02 (0.09) |
| Calm | 5.36 | 1.42 | 1.00 | 7.00 | -0.92 (0.06) | 0.58 (0.06) |
| Contented | 4.90 | 1.71 | 1.00 | 7.00 | -0.83 (0.07) | -0.11 (0.07) |
| Relaxed | 4.89 | 1.62 | 1.00 | 7.00 | -0.58 (0.06) | -0.33 (0.06) |
| StatementA | 5.15 | 1.47 | 1.00 | 7.00 | -0.84 (0.06) | 0.42 (0.06) |
| StatementB | 4.67 | 1.66 | 1.00 | 7.00 | -0.42 (0.07) | -0.62 (0.07) |
| Plausibility in Contradiction | 0.56 | 0.50 | 0.00 | 1.00 | - | - |

*Note*: i = item from the locus of attention subscale of the AHS; M = Mean; SD = standard deviation; Min = Minimum; Max = Maximum; s.e. = standard error from the Skew and Kurtosis statistics.

**Table 4. Comparison of goodness-of-fit indices across models.**

| Model | $\chi^2$ | Df | CFI | TLI | RMSEA (95% CI) | SRMR |
|---|---|---|---|---|---|---|
| Measurement model | 12.12 | 8 | 0.99 | 0.99 | 0.029 (0.001, 0.060) | .020 |
| Structural model | 61.29 | 19 | 0.93 | 0.90 | 0.073 (0.053, 0.094) | .048 |

*Note*: TS model: Latent model for Thinking Style; Structural model: SEM model for Thinking Style on Plausibility in Contradiction, mediated through Mixed Emotions; χ2 = Chi-square; df = degrees of freedom; CFI = Comparative Fit Index; TLI = Tucker-Lewis Index; RMSEA = Root Mean Square Error of Approximation; SRMR = Standardized Root Mean Residual.

contradiction as the outcome variable, thinking style as the predictor variable, and mixed emotions as the mediator. We obtained an adequate goodness of fit for this model: $\chi^2$ (19) = 61.29, CFI = .93, TLI = .90, RMSEA (95%) = 0.073 (0.053, 0.094) (see Table 4). Although the goodness of fit was slightly below Hu & Bentler's cut-offs [44], we decided to keep the model as adding more residual correlations between the observed variables would not improve more the model. The direct path from thinking style to plausibility in contradiction was significant, $B = 0.32$, $SE = 0.06$, $p < .001$, indicating once again that holistic individuals dealt with contradicting statements by considering them as more plausible than analytic individuals. The path from thinking style to mixed emotions, the mediator, was also significant, $B = -0.42$, $SE = 0.11$, $p < .001$, indicating that holistic individuals experienced more mixed emotions than analytic individuals did. Finally, the path from mixed emotions to plausibility in contradiction was significant, $B = -0.18$, $SE = 0.03$, $p < .001$, indicating that individuals who experienced more mixed emotions also found more plausibility in contradictions. The remaining indirect path, from thinking style to plausibility in contradiction, via the experience of mixed emotions, was found to be significant, $B = 0.07$, $SE = 0.02$, $p = .001$, supporting the mediating role for mixed

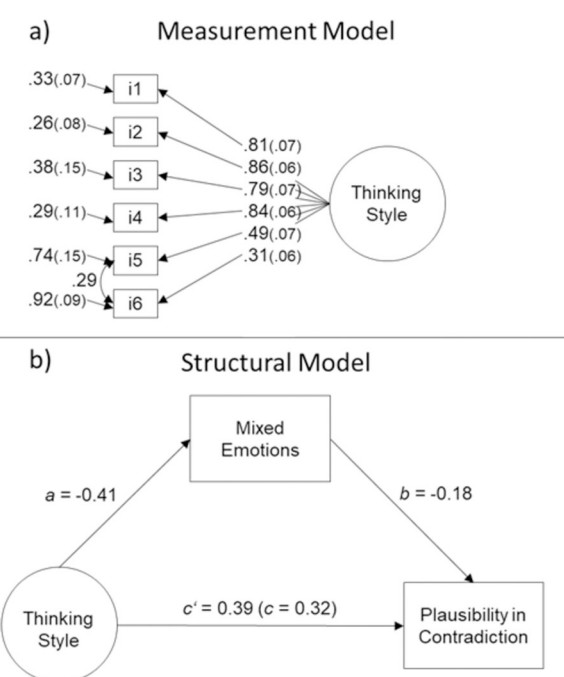

**Fig 2.** (a) top panel: Item factor loadings, Study 2; (b) bottom panel: Final structural equation model; standardized path coefficients are presented, Study 2. $^*p < 0.001$. Total effect and direct effect (in brackets) are reported.

emotions in the relationship between thinking style and plausibility in contradiction (see Fig 2B). The partially standardized indirect effect was 0.08 [56].

As in Study 1, when we used the sum of each of the scores for both statements to create one overall index of plausibility in contradictory information, the results were similar and the final structural model including the mediation fitted: $\chi^2$ (18) = 74.54, CFI = .95, TLI = .92, RMSEA (95% CI) = .071 (.057, .087).

## Discussion

The results of Study 2 were consistent with Study 1. That is, participants with a holistic thinking style dealt with the contradictory statements by considering them as more plausible than participants with an analytic thinking style. As predicted, the relationship between thinking style and the strategy to deal with contradiction was mediated by the experience of mixed emotions. Specifically, and consistent with previous research [22,23], participants with a holistic thinking style experienced more mixed emotions than participants with an analytic thinking style, and experiencing more mixed emotions was associated with more plausibility in contradictions.

## General discussion

Across two studies, we showed that participants with a holistic thinking style dealt with contradictory statements by considering them as more plausible than participants with an analytic thinking style (Study 1), and that the relationship between thinking style and plausibility in contradiction was mediated by the experience of mixed emotions (Study 2). Interestingly, although holistic-analytic thinking style is culturally embedded [8], our study suggests that it can also be used as an individual-difference variable to distinguish between holistic and analytic thinkers within the same culture [57] regarding the way people deal with contradictions. Our research captured this idea using a structural equation modeling approach, which increased the internal validity of the studies and added value over previous literature using regression analyses to study thinking style [57–59].

Previous research has shown that individuals in Western cultures usually report feeling emotions that are either positive or negative, showing a strong and negative correlation between positive and negative emotions [22,23]. However, individuals in Eastern cultures show no correlation or even a positive correlation between positive and negative emotions, indicating that they may feel emotions of both valence at the same time [26,33]. In line with previous research, we demonstrate that this is also true when we use holistic-analytic thinking style as an individual-difference variable instead of focusing on cultural differences [55]. Importantly, another contribution of the present work resides in the mediating role given to mixed emotions. Specifically, Study 2 showed that having mixed emotions is associated with finding contradictions as more plausible.

There are individual variables that could further modify the effects uncovered in this research. For instance, individuals may differ in how certain they are about their traits [60–62]. Those who are more certain about their thinking style should show the effect to a greater extent (e.g., "if I am certain that I have a holistic style, I would perceive contradictions to an even greater extent"). In contrast, those who doubt their thinking style would be expected to show a reduced pattern (e.g., "I doubt that I have a holistic style, so I do not know if I could perceive contradictions"). Therefore, future research should explore whether certainty measures can enhance the predictive power of holistic-analytic thinking style on confrontation of contradictions.

Another element that could affect the results of this research is the timing in which the feelings are measured. In our research, the path analysis followed this order: first the thinking

style, then the feeling of mixed emotions, and finally the plausibility in contradictions. If we change the order such that participants find the contradiction first then report how they feel, we may find a different potential psychological mechanism, namely psychological discomfort. Previous research has shown that holistic individuals feel less discomfort when they encounter contradictions than analytic individuals [9,18,21]. Therefore, future research might include a measure of psychological discomfort *after* the contradiction is presented. This can be also very informative regarding the timing in which the feelings are introduced (see [63,64], for reviews on the importance of timing). These two mechanisms are not incompatible with one another. Rather, which mechanism is activated depends on the timing in the procedure during which feelings are elicited or recorded. Future research should also explore whether the psychological mechanism (mixed emotions experience) proposed to explain the relationship between thinking style and contradiction is capable of explaining other relationships between thinking style and different cognitive outcomes such as categorization [33], biases and heuristics [65,66], consumer behavior [67,68], and many others.

This study has some limitations that must be considered. First, although we use a well-established scenario for testing the dealing with contradiction, future studies should be undertaken to extend and generalize the findings to other contexts with contradictory information (e.g., cognitive dissonance scenarios, or moral dilemmas). Second, this study was based on participants' self-report measures. Future studies could manipulate thinking style [40,41] and mixed emotions [30,69,70] in order to capture the explanatory nature of the phenomenon. Third, regarding the thinking style measure, instead of including the full Analysis-Holism Scale we used one of its subscales (i.e., locus of attention). Since the time and space to run the present studies were limited we chose a costless measure without sacrificing reliability or validity. Future studies should also use a more complete measure of thinking style or a different procedure to assess them, such as the framed line test or the triad task (for a review, see [57]). Fourth, regarding the variable mixed emotions, someone might wonder what is the role of the intensity of mixed emotions. In this research, we have focused on the occurrence of mixed emotions but not on the degree to which people experience this emotional state. This is an interesting question that is worth exploring in further studies, either by manipulating or measuring experiences of mixed emotions and their intensity. Finally, we acknowledge that a design with an experimental approach would provide evidence for causal mediation [71,72]. In fact, the correlational nature of our design prevents us to conclude whether there is a model with more explanatory power than the proposed one.

In conclusion, the current research provides an important extension to prior work on thinking style and the confrontation strategies of contradiction. Specifically, the present studies extend the extant literature in two important ways. First, this research specified the experience of mixed emotions as a potential mechanism driving the relationship between thinking style and the confrontation of contradiction. Second, the individual-differences approach we used is different than the cultural-differences approach followed by Peng and Nisbett [9] and Spencer-Rodgers et al. [73]. In their research, they showed that Eastern individuals reconcile more contradiction and have more mixed emotions than Western individuals. We extended that research by showing that individuals who adopt a holistic thinking style are better able to adopt a compromise approach toward contradictions and have more mixed emotions than analytic individuals.

## Acknowledgments

We would like to thank Dr. Dilney Gonçalves, Dra. María Cantero, Dr. David Moreno, and Dr. Miguel Sorrel for their comments on an early version of the first draft and review.

## Author Contributions

**Conceptualization:** David Santos, Blanca Requero.

**Formal analysis:** Manuel Martín-Fernández.

**Funding acquisition:** David Santos.

**Methodology:** David Santos, Blanca Requero, Manuel Martín-Fernández.

**Project administration:** David Santos.

**Resources:** David Santos.

**Supervision:** David Santos.

**Validation:** David Santos.

**Visualization:** Manuel Martín-Fernández.

**Writing – original draft:** David Santos, Blanca Requero, Manuel Martín-Fernández.

**Writing – review & editing:** David Santos, Blanca Requero, Manuel Martín-Fernández.

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
