## [Decision Letter · Decision Letter 0]

4 Jun 2021

PONE-D-21-07959

Individual differences in thinking style and dealing with contradiction:The mediating role of mixed emotions

PLOS ONE

Dear Dr. Velasco,

Thank you for submitting your manuscript to PLOS ONE. After careful consideration, we feel that it has merit but does not fully meet PLOS ONE’s publication criteria as it currently stands. Therefore, we invite you to submit a revised version of the manuscript that addresses the points raised during the review process.

I appreciated reading this paper that provides valuable information regarding thinking styles, emotions, and contradictory information. All reviewers note strengths of the work and problems that would need to be corrected. All the different issues should be addressable.

We look forward to receiving your revised manuscript.

Kind regards,

Gaëtan Merlhiot

Academic Editor

PLOS ONE

Journal Requirements:

2. Thank you for including your ethics statement:  "IE Research Committee (Number IERC/39-2019-2020). Written informed consent was obtained. ".   

1. Please amend your current ethics statement to include the *full* name of the ethics committee/institutional review board(s) that approved your specific study.

2. Please amend your current ethics statement to confirm that your named institutional review board or ethics committee specifically approved this study.

Reviewers' comments:

Reviewer's Responses to Questions

**Comments to the Author**

1. Is the manuscript technically sound, and do the data support the conclusions?

Reviewer #1: Partly

Reviewer #2: Partly

2. Has the statistical analysis been performed appropriately and rigorously? 

Reviewer #1: No

Reviewer #2: Yes

3. Have the authors made all data underlying the findings in their manuscript fully available?

Reviewer #1: No

Reviewer #2: Yes

4. Is the manuscript presented in an intelligible fashion and written in standard English?

Reviewer #1: Yes

Reviewer #2: Yes

5. Review Comments to the Author

Reviewer #1: Review of “Individual differences in thinking style and dealing with contradiction: The mediating role of mixed emotions”

This paper tests the well-conceived idea that individual differences in analytic/holistic thinking predict how people deal with contradictory thoughts (mirroring cross-cultural work). I think the article is well-written and meets most of PLOS ONE’s publication criteria. However, I have some concerns about the analytic approach used by the authors, which I think either needs to be changed or much better explained and justified. Otherwise, I think the paper is strong—it has a plausible hypothesis that is connected to previous literature and testable with the present measures.

1. The authors use the Locus of Attention subscale from the Analysis-Holism Scale, which made me wonder why the Attitudes Toward Contradiction subscale isn’t the more intuitively appropriate subscale? Isn’t that the subscale specifically about accepting contradictions? I think the choice of subscale at least needs to be explained in more detail, and perhaps the limitations of this particular subscale choice could be elaborated upon in the discussion (depending on the authors’ reasons for the choice).

2. I think it would be useful to explain and justify the choice of SEM (rather than the more intuitive correlation and regression approaches) when it is first introduced. The authors briefly touch on this approach in the general discussion, but I think readers will wonder why they are using it earlier on. Related to that, because there is a more intuitive, simple approach, I would be inclined report the correlations and regressions that would normally be used (at least in the supplemental material, if the authors do not feel it is appropriate in the main text). Presumably these show similar patterns of results.

3. I think the authors should report descriptive statistics for their measures, as well as correlations among them. This is important for the reader, and relates to some concerns about their analytic approach:

3a. My primary concern is about the measurement of plausibility contradictions. I do not think it is analytically appropriate to lose the continuous the data the authors have collected in favor of creating a binary variable and discarding of a relatively large percentage of participants. Scoring continuous measures into categories reduces power and is generally less informative than keeping the continuous measures (and after all, why were continuous measures used if categories were desired). From a theoretical standpoint, the coding also obscures important differences. Surely someone who rates statement 1 as a 5 and statement 2 as a 7 on plausibility is showing less comfort with contradiction than someone who rates statement 1 as a 5 and statement 2 as a 5? Yet they would get the same score. Similarly, rating statement 1 as a 7 and rating statement 2 as 1 would get you the same score as rating statement 1 as a 5 and statement 2 as a 3, but surely there is more tolerance for contradiction in the latter case.

3b. Perhaps a more appropriate approach might be to see whether thinking style moderates the relation between the two plausibility ratings. The ratings should presumably be more negatively correlated for people higher in analytic thinking (if seeing one as more plausible means seeing the other as less plausible), but less negatively correlated for people higher in holistic thinking. That would allow the authors to preserve all of their data and use their continuous measures.

3c. A tricky aspect of these data, for both the plausibility ratings and the emotion ratings, is that sometimes small differences clearly indicate contradictions and sometimes it’s less clear that they do. With the emotions, for example, someone with a difference score of 0 could be low on both positive and negative emotions, or they could be high on both (or in the middle, of course). Being high on both does seem to clearly be a mixed emotional state. But being low on both might not necessarily be a mixed state, right? If a participant is not feeling any of the given emotions strongly, that does not necessarily mean they tend to experience more mixed emotions generally, because it just means they are not, in the moment, experiencing any of the provided emotions. This is also partly why I think the descriptive data are so important, as well as perhaps figure(s) (e.g., a graph with thinking style on the x-axis and two lines, one for each plausibility statement rating). Where are the data concentrated? How do we interpret different ends of the measures of plausibility and emotion? This is a tricky issue, and I think the authors need to address it and its complexity in the paper.

4. A more minor point, but how were the sample sizes determined? Did the authors conduct power analyses, or use a different strategy?

5. When I tried to access the data on OSF, I did not have access. Please ensure the OSF page is open, or include the anonymous reviewer link instead.

In sum, I think the authors are proposing an interesting and very worthwhile idea and have data that could potential be used to test it, but I am not sure the current analytic approach is optimal. With a revised approach, I would be excited to see this paper progress toward publication.

Reviewer #2: Review of “Individual differences in thinking style and dealing with contradiction:The mediating role of mixed emotions

The authors set out to determine whether the thinking style is associated with the way people deal with contradictory information and experiencing mixed emotions can explain this association. I found this study truly interesting. However, I do have some major issues with the data analysis and some additional minor suggestions for improvement.

1) I think the most major limitation of the paper is that it lacks valid instruments to measure holistic thinking. The authors measured thinking style based on a self-report measure instead of a Triad Task performance-based measure. But this is a limitation, which I believe requires further discussion in the limitations part.

2) The second limitation of the paper is that it lacks an overarching definition of holistic thinking. It’s clear that holism is a multifaceted concept that consists of several dimensions, but we don’t know the exact components of holistic thinking besides dialectical thinking and contradiction. If I propose intelligence is a multi-dimensional concept, I should provide sufficient information about its subcomponents.

3) Since all studies are correlational, they are not well suited for testing mediation. First of all, the reverse causal direction is possible. As Lemmer and Gollwitzer (2017; https://doi.org/10.1016/j.jesp.2016.05.002) showed, testing reverse alternative models can also lead to mistaken results. You cannot conclude in a correlational design whether there is a better model since in psychology most variables (except certain demographics like gender or age) cannot be measured perfectly reliably. This kind of mediation analysis would only work when the IV is experimentally manipulated. In a correlational design, instead of mediation analysis, a better statistical test is to conduct a stepwise regression and investigate how much variance in the IV to DV relationship is accounted for by adding alternative mediators to the model. I know in the past, it had widespread use in psychology literature as an “individual differences approach,” but the authors need to acknowledge this fact more in the limitations part. More importantly, the authors should avoid using causal language throughout the manuscript (such as “may affect” in the “Abstract” and the “Overview of the present research”). Similarly, there are no independent and dependent variables in this study since there was no experimental manipulation. Please label them as the predictors and the outcome variables throughout the manuscript (such as P10 “Statistical analysis”).

4) How did you determine the required sample sizes? Is there any stopping rule?

Overall, I think (1) the authors should provide a conceptual explanation of what holism is and (2) (a) either discuss the misfit of the term “mediation” in the correlational design or (b) conduct experimental research by manipulating holistic vs. analytic thinking as in Talhelm et al. (2015 https://doi.org/10.1177/0146167214563672) to test for causal mediation.

I always sign my reviews,

Onurcan Yilmaz

6. PLOS authors have the option to publish the peer review history of their article (what does this mean?). If published, this will include your full peer review and any attached files.

Reviewer #1: No

Reviewer #2: No

---

## [Author Response · Author response to Decision Letter 0]

1 Jul 2021

July 1st, 2021

Dr. Gaëtan Merlhiot,

Academic Editor

PLOS ONE

Dear Dr. Merlhiot,

We would like to express our sincere gratitude to you and the reviewers for the helpful and insightful feedback provided on June 4th 2021 for our manuscript entitled “Individual differences in thinking style and dealing with contradiction: The mediating role of mixed emotions” (#PONE-D-21-07959). We were glad to see that our manuscript has merit. 

Please, find below our responses to the concerns and comments raised by the reviewers. Please do not hesitate to contact us if you have any additional thoughts, questions, or queries concerning this revised manuscript. Thank you once more for inviting us to revise and resubmit our work. We believe that the revised manuscript is significantly improved and can make a more complete contribution to PLOS ONE than the previous version. 

Responses to Reviewer 1: 

Reviewer 1 provided a positive response to our manuscript and we very much appreciate it. This reviewer began by saying that our “article is well-written and meets most of PLOS ONE’s publication criteria” and that our “paper is strong—it has a plausible hypothesis that is connected to previous literature and testable with the present measures.” At the same time, Reviewer 1 raised a number of issues that we address below:

1. Justification of the Locus of Attention subscale: Reviewer 1 wondered why this subscale has been selected instead of the Attitudes Toward Contradiction subscale of the AHS.

We thank the reviewer for noticing this point and allowing us to clarify why this specific subscale has been chosen. In fact, the extent of specificity between the subscale of attitudes towards contradiction and the dependent variable (plausibility in contradictions) may be higher considering that this measure focuses on how people deal with two opposite propositions (an example item, “It is desirable to be in harmony, rather than in discord, with others of different opinions than one’s own”). However, we consider that this subscale is less suitable for capturing the variability of our proposed mediator (mixed emotions) insofar as it has nothing to do with how to deal with disagreements in points of view. That is why we chose a subscale with a broader focus that reflects the ability to maintain discrepant elements (whether emotions or opinions) simultaneously within the same reality.

According to the locus of attention subscale, holistic thinkers focus more on how the multiple components come together as part of a coherent overall experience compared to analytic processors who focus more on isolated components (an example item, “The whole, rather than its parts, should be considered in order to understand a phenomenon”). Although the specificity of this subscale might be lower with respect to the DV, we chose it because we consider that it is capable of encompassing mixed emotions and plausibility of the contradiction equally.

Furthermore, both scales keep a high correlation (r = .65; see Choi et al., 2007) as they are dimensions of the construct "Holistic-analytic thinking style" (regarding this last mention, we have expanded more information in the point #2, Reviewer 2). Someone might question whether it would not have been more appropriate to use the full scale with all its components, so this concern has been included in the part of limitations (page 23, lines 9-14 “Third, regarding the thinking style measure, instead of including the full Analysis-Holism Scale we used one of its subscales (i.e., locus of attention). Since the time and space to run the present studies were limited, we chose a costless measure without sacrificing reliability or validity. Future studies should also use a more complete measure of thinking style.”).

2. Justification of the data analysis technique (SEM): Reviewer 1 noted that the use of the technique (SEM) should be explained over the use of regression analysis.

We would like to thank the reviewer for this comment. In contrast with classic linear models (e.g., ANOVA, linear regression, generalized linear regression), SEM models allow researchers to control the measurement error, something crucial when dealing with latent variables, as it is the case of the holistic-analytic thinking style continuum. 

We agree that a better justification for the analytical approach could be included in the manuscript. We have added a brief description of this approach in the statistical analyses section of the first study (page 11, paragraph 2).

3. Inclusion of descriptive statistics and correlations: Reviewer 1 asked us to include correlations and descriptive statistics of the measures.

We appreciate the comment and the possibility of including more information regarding our measures. We have added a brief descriptive analysis of the variables included in each study (see page 12, and pages 18-19), including the correlations between the statements in the first study, and between the statements and mixed emotions in the second. 

3. a. Measurement of the variable Plausibility of Contradictions: Reviewer 1 wondered why we decided to dichotomize the criterion variable instead of using a continuous approach (e.g., summing up the two statements).

We thank the reviewer for the opportunity to explain our instantiation of the criterion variable. Taking the variable as a composite measure of the statements (i.e., summing or averaging the two statements) might lead to some problems with the conceptualization of the plausibility of contradictions. Please allow us to briefly explain our rationale. Imagine these 2 participants: Participant A considers the plausibility of both statements to be a 5 (out of 7), whereas participant B considered the plausibility of the first statement to be a 7 and the second statement to be a 3. Once we compute the composite index for each participant, both participants A and B receive the same score (i.e., 10). However, in this particular case, participant A showed a response pattern more consistent with the plausibility of contradictory information than participant B. Thus, we think that the continuous method of scoring the DV does not distinguish between participants in terms of how much they compromise when dealing with contradictory information, at least when it comes to participants who provide responses around the mid-point.

We decided, however, to report this other way of computing the criterion variable using the sum of the two statements as an overall indicator of plausibility in contradictory information (see page 14, paragraph 2, and page 20, paragraph 2), such that higher scores indicated that the two statements were perceived as highly plausible and lower scores indicated that one of the statements was perceived as less plausible that the other. When using this way of scoring the criterion variable, the data fitted in Study 1, χ2 (13) = 35.81, CFI = .97, TLI = .95, RMSEA (95% CI) = .059 (.041, .077), and Study 2, χ2 (18) = 74.54, CFI = .95, TLI = .92, RMSEA (95% CI) = .071 (.057, .087). This analyzes were already included in the previous version of the manuscript.

3. b. Different analytical approach: Reviewer 1 suggested to model whether thinking style moderates the relation between the two plausibility statements.

In point #2 above, we have explained why we have chosen a SEM approach rather than treating the variables as observed and using a regression approach. Nonetheless, we have modeled our results by using the approach the reviewer proposed and the results are consistent with our findings. In this regression approach, we used each statement separately and continuously. Thus, Statement 1, Thinking Style (the mean of the 6 items), and the interaction term (i.e., Statement 1 × Thinking Style) were used as predictors, and Statement 2 was used as the criterion variable. The criterion variable (Statement 2) was submitted to a multiple regression analysis. The key two-way interaction was computed using the PROCESS macro for SPSS (model 1; Hayes, 2013). The continuous variables (i.e., Statement 1 and Thinking Style) were mean-centered to reduce multi-collinearity issues when computing interaction terms. The criterion variable (i.e., Statement 2) was regressed onto the predictors (Statement 1 and Thinking Style) as well as their interaction term using a hierarchical regression (i.e., main effects in the first step, followed by two-way interaction). Based on Cohen and Cohen (1983), all main effects and interactions were interpreted in the first block in which they are shown in the regression analyses.

The regression analysis revealed a main effect of statement 1 on statement 2, B = -0.280, t(682) = -6.751, p < .001, indicating that the more plausible individuals perceive statement 1, the less plausible they perceive statement 2 and vice versa.

Interestingly, this main effect was qualified by a significant interaction between statement 1 and thinking style, B = 0.074, t(681) = 2.047, p = .041. As illustrated in Figure 1, among those with lower AHS scores (-1SD; people who tend to be more analytic), statement 1 was negatively related to statement 2, B = -0.347, t(681) = -6.573, p < .001. For those with higher AHS scores (+1SD; people who tend to be more holistic), statement 1 was also negatively related to statement 2, B = -0.201, t(681) = -3.552, p = .0004, but this negative correlation was significantly less stronger as indicated by the significant two-way interaction (thus indicating greater plausibility in contradiction). 

This regression approach yielded very similar results compared to our SEM approach. That is, a higher plausibility of contradiction among individuals who tend to be more (vs. less) holistic. 

In Study 2, the findings were less robust although the pattern of results was replicated. The relationship between Statement 1 and Statement 2 as a function of Thinking Style approached significance, B = 0.063, t(620) = 1.775, p = .076.

Figure 1. Relationship between Statement 1 and Statement 2 as a function of Thinking Style.

We provided these analyses to show the reviewer that both procedures lead to similar conclusions. Although we agree with the reviewer that modeling the interaction between the statements could be interesting, we do not deem necessary to add it to the manuscript as it is somewhat redundant with the analyses currently reported in the paper. 

In addition, modelling this type of moderations in SEM is tricky, as one of the predictors is a latent variable (Thinking Style) and the other predictor is an observed variable of just one item (Statement 1). Many authors have argued that this moderation in SEM can produce problems of convergence of the model (Foldnes & Hagtvet, 2014). However, we still think that our SEM approach provides more information than the regression approach for the reasons explained in point #2.

Cohen J, Cohen P, West SG, Aiken LS. Applied multiple regression. Correlation Analysis for the Behavioral Sciences. 1983;2.

Foldnes N, Hagtvet KA. The choice of product indicators in latent variable interaction models: Post hoc analyses. Psychol Methods. 2014 Sep;19(3):444-57. 

Hayes AF. Introduction to mediation, moderation, and conditional process analysis: A regression-based approach. Guilford publications; 2017 Dec 13.

3. c. Measurement of the variable Mixed Emotions: Reviewer 1 was concerned about the measurement of mixed emotions and that we did not take into account the intensity of the mixed state. Also, this reviewer asked us to include descriptive data or a graph showing where the data are concentrated. 

This is a fair comment. The research question proposed in this work is whether the occurrence of mixed emotions can serve as a mediator mechanism between thinking style and plausibility in contradictions regardless of the intensity of the emotions. To test this hypothesis, we used an operationalization of mixed emotions following previous research (Hamamura et al., 2008; Hui et al., 2009), and as the reviewer points out, this does not capture the intensity of the emotions experienced. Given that it seems to us an interesting question, we have included in the discussion the proposal of considering the role of the intensity of mixed emotions in subsequent studies (see limitations and future research part, page 23, lines 15-19). Furthermore, since we do not want to mislead the reader on this aspect, we have deleted the mention of intensity in the method section (in such a way that in the new manuscript it appears only in the discussion). We thank Reviewer 1 for this suggestion that raises interesting lines of follow-up research to this one.

[…] Forth, regarding the variable mixed emotions, someone might wonder what is the role of the intensity of mixed emotions. In this research, we have focused on the occurrence of mixed emotions but not on the degree to which people experience this emotional state. This is an interesting question that is worth exploring in further studies, either by manipulating or measuring experiences of mixed emotions and their intensity. […]

4. Sample size: Reviewer 1 asked us what strategy we used to determine the sample size. 

Sample size was determined following the recommendations of several papers on the topic. In SEM, the statistical power depends on model complexity, the estimation method, and the response format of the observed variables (i.e., items). Hence, there are no clear rules of thumb to determine the minimum sample size. For a 3-factor correlated model with categorical data, a sample size of 400-500 participants is needed to ensure acceptable Type I and Type II error rates (Bandalos, 2014; Forero et al., 2009; Wolf et al, 2013). In general, when using a SEM approach, the higher the sample size, the better. 

The sample size of both studies is above these minimum requirement of participants to ensure enough statistical power. 

Bandalos DL. Relative performance of categorical diagonally weighted least squares and robust maximum likelihood estimation. Struct Equ Modeling. 2014;21(1):102–16.

Forero CG, Maydeu-Olivares A, Gallardo-Pujol D. Factor analysis with ordinal indicators: A Monte Carlo study comparing DWLS and ULS estimation. Struct Equ Modeling. 2009;16(4):625–41.

Wolf EJ, Harrington KM, Clark SL, Miller MW. Sample size requirements for structural equation models: An evaluation of power, bias, and solution propriety: An evaluation of power, bias, and solution propriety. Educ Psychol Meas. 2013;76(6):913–34.

5. Open data: Reviewer 1 could not access the data on OSF. 

 Thank you for noticing this issue. We have granted access to the link to anyone but anonymized for blind review purposes. The new link that we have included in the manuscript is: https://osf.io/yadft/?view_only=f1f8cefed3f144a9bd7f8d3a888dc476

Responses to Reviewer 2: 

Reviewer 2 expressed that “I found this study truly interesting.” We really appreciate this positive comment and the feedback of their review. Furthermore, this Reviewer brought up additional comments that we address below:

1. Measurement of Holistic-Analytic Thinking Style: Reviewer 2 expressed that there were other procedures to measure holistic-analytic thinking style that could have used. At the same time, this reviewer was concerned about the validity of the scale we used to measure thinking styles.

In the present research, we have opted to measure the key construct by using a self-reported measure (i.e., the Locus of Attention subscale from the AHS). As the reviewer points out, there exist multiple procedures to measure holistic-analytic thinking (for a review, see Na et al., 2020). The original work by Choi et al. (2007) demonstrates that the AHS is both a reliable and valid instrument to measure systematic cognitive differences in holistic-analytic thinking. Specifically, this measure has been shown to predict performance on two cognitive tasks (categorization and causal reasoning). Nonetheless, we now acknowledge in the general discussion that future research could not only manipulate thinking styles but also measure them with different procedures, such as the framed line test or the triad task (for a review, see Na et al., 2020). 

Na J, Grossmann I, Varnum ME, Karasawa M, Cho Y, Kitayama S, Nisbett RE. Culture and personality revisited: Behavioral profiles and within‐person stability in interdependent (vs. independent) social orientation and holistic (vs. analytic) cognitive style. J of Pers. 2020 Oct;88(5):908-24.

2. Overarching definition of holistic thinking: Reviewer 2 asked us to extend our conceptualization of holistic-analytic thinking style.

We have followed this recommendation and the revised manuscript now includes a paragraph that addresses the different components that define holistic-analytic thinking style. These new lines reflect the multifaceted nature of the construct by explicitly mentioning and describing its dimensions (page 3, second paragraph).

 […] These theories define holistic-analytic thinking style as a multifaceted construct composed by several dimensions. The first dimension is causality, which considers the presence of complex causalities and the elements of the universe as interconnected and interrelated (vs. the universe consists of elements that are independent of each other, Koo & Choi, 2005). The second one is attitudes towards contradiction, which assesses the preference for resolving the contradiction through a reconciliation strategy, seeking the “middle way” between opposing propositions (vs. contradictions are resolved by choosing one of the two opposite propositions, Spencer-Rodgers et al., 2010). The third, perception of change, refers to the tendency to perceive the elements as being in constant change and unpredictable (vs. linear changes and predictable, Yama & Zakaria, 2019). The last dimension is locus of attention, which places the focus on “the big picture”, considering the elements of the stimulus as a whole (rather than decomposing the stimulus in their parts ignoring the context, Nisbett & Miyamoto, 2005). […]

Koo M, Choi I. Becoming a holistic thinker: training effect of oriental medicine on reasoning. Pers Soc Psychol Bull. 2005;31(9):1264–72.

Nisbett RE, Miyamoto Y. The influence of culture: holistic versus analytic perception. Trends Cogn Sci. 2005;9(10):467–73.

Spencer-Rodgers J, Williams MJ, Kaiping Peng. Cultural differences in expectations of change and tolerance for contradiction: a decade of empirical research. Pers Soc Psychol Rev. 2010;14(3):296–312.

Yama H, Zakaria N. Explanations for cultural differences in thinking: Easterners’ dialectical thinking and Westerners’ linear thinking. J Cogn Psychol (Hove). 2019;31(4):487–506.

3. Mediation model: Reviewer 2 was concerned about the reverse-causality problem in correlational designs. Also, this reviewer questioned wording indicating a more causal role of the variables.

We agree with Reviewer 2 about the correlational nature of our mediation. We also agree that the causality implied by some sentences is not warranted by the design of our study. Therefore, we now use more accurate wording to express the correlational nature of this mediation model throughout the manuscript.

Finally, in addition to the changes in wording, we have considered important to clarify further this issue in the limitations section of the manuscript: “Finally, we acknowledge that a design with an experimental approach would provide evidence for causal mediation [71, 72]. In fact, the correlational nature of our design prevents us to conclude whether there is a model with more explanatory power than the one proposed.” (page 23, lines 20-23). 

71. Fiedler K, Harris C, Schott M. Unwarranted inferences from statistical mediation tests–An analysis of articles published in 2015. J Exp Soc Psychol. 2018 Mar 1;75:95-102.

72. Lemmer G, Gollwitzer M. The “true” indirect effect won't (always) stand up: When and why reverse mediation testing fails. J Exp Soc Psychol. 2017 Mar 1;69:144-9.

4. Sample size: Similarly to point #4 of Reviewer 1, this Reviewer also asked us what strategy we used to determine the sample size and whether there was a stopping rule. 

We aimed for a sample size above 500 participants (see response to point #4 of Reviewer 1). Given that the on-line sampling strategy have some tradeoffs (is cost-efficient, but it is necessary to control the quality of the answers given by the participants), we decided to stop the data collection when the sample size approached N = 650. We opted for this number in case some participants failed the attention checks of the study and need to be removed from the study (something that only could be assessed after the data collection). 

Summary

We hope that this letter clarifies how we have addressed the key issues raised during the review process. Some of the issues were addressed by attempting to clarify our writing, and others were addressed with new analysis of the data. The feedback from you and both reviewers has certainly helped us to improve the manuscript quality considerably. Because the reviewers seemed positively disposed toward the potential contribution of the paper in their reviews, as did you, we hope that you find this version –which addresses the concerns mentioned during the review process - suitable for publication. Please do not hesitate to contact us if you have any additional query.

Best regards,

The authors.

---

## [Decision Letter · Decision Letter 1]

31 Aug 2021

PONE-D-21-07959R1

Individual differences in thinking style and dealing with contradiction: The mediating role of mixed emotions

PLOS ONE

Dear Dr. Velasco,

Thank you for submitting your manuscript to PLOS ONE. After careful consideration, we feel that it has merit but does not fully meet PLOS ONE’s publication criteria as it currently stands. Therefore, we invite you to submit a revised version of the manuscript that addresses the points raised during the review process.

All comments have been considered by the authors with clear explanations. Please take the last remaining comments into account. 

We look forward to receiving your revised manuscript.

Kind regards,

Gaëtan Merlhiot

Academic Editor

PLOS ONE

Journal Requirements:

Additional Editor Comments (if provided):

Reviewers' comments:

Reviewer's Responses to Questions

**Comments to the Author**

1. If the authors have adequately addressed your comments raised in a previous round of review and you feel that this manuscript is now acceptable for publication, you may indicate that here to bypass the “Comments to the Author” section, enter your conflict of interest statement in the “Confidential to Editor” section, and submit your "Accept" recommendation.

Reviewer #1: All comments have been addressed

Reviewer #2: All comments have been addressed

2. Is the manuscript technically sound, and do the data support the conclusions?

Reviewer #1: Yes

Reviewer #2: Yes

3. Has the statistical analysis been performed appropriately and rigorously? 

Reviewer #1: Yes

Reviewer #2: Yes

4. Have the authors made all data underlying the findings in their manuscript fully available?

Reviewer #1: Yes

Reviewer #2: Yes

5. Is the manuscript presented in an intelligible fashion and written in standard English?

Reviewer #1: Yes

Reviewer #2: Yes

6. Review Comments to the Author

Reviewer #1: The authors have appropriately addressed my concerns. Although I might have liked to see the newly conducted regression (producing the expected interaction) in the supplemental material, the authors' justification of the omission of this analysis is appropriate. I congratulate the authors on a nice paper that will contribute to the literature.

Reviewer #2: The authors addressed most of my comments. The only point left unaddressed is the use of causal language. Although the authors acknowledged the fact that the design is correlational and no causal language should be used, they have a title called "Dependent Variable", and there were other uses of the same frame in the manuscript. However, in a correlational design where no manipulation exists, there is not any IV or DV. So I suggest avoiding such a language. Predictor vs. Outcome distinction is the right one in a correlational design.

I always sign my reviews,

Onurcan Yilmaz

7. PLOS authors have the option to publish the peer review history of their article (what does this mean?). If published, this will include your full peer review and any attached files.

Reviewer #1: No

Reviewer #2: **Yes: **Onurcan Yilmaz

---

## [Author Response · Author response to Decision Letter 1]

6 Sep 2021

September 6th, 2021

Dr. Gaëtan Merlhiot,

Academic Editor

PLOS ONE

Dear Dr. Merlhiot,

We are very thankful for the opportunity to resubmit our work entitled “Individual differences in thinking style and dealing with contradiction: The mediating role of mixed emotions” (#PONE-D-21-07959) to PLOS ONE. We are delighted to see that we have successfully addressed the comments and concerns of the reviewers in the first round and that they were very enthusiastic about the contribution of our work.

In this second round, we have addressed the a suggestion raised by the Reviewer 1. 

Response to Reviewer 1: 

Reviewer 1 noted that we have addressed all of his/her concerns and considers this a good paper with potential contribution to the literature. We are really happy to hear these encouraging comments. 

Response to Reviewer 2:

We would like to thank Reviewer 2 for his appreciation of our changes in the revised version of the manuscript. He (self-identified reviewer) only raised a minor issue regarding the appropriate use of variable labels according to a correlational design. We agree that the correct use of the labels should be Predictor and Outcome instead of Independent and Dependent Variables. Therefore, we have modified the terms to conform to the language of correlational designs throughout the manuscript. We thank Onurcan Yilmaz for his helpful feedback along the process. 

Summary

We hope this letter clarifies how we have addressed the few remaining issues raised during this second review round. Furthermore, we would like to sincerely thank reviewers for their time and constructive feedback during the entire review process which has helped us to considerably improve the paper. Please do not hesitate to contact us if you have any additional query.

Best regards,

The authors

---

## [Editor Report · Decision Letter 2]

14 Sep 2021

Individual differences in thinking style and dealing with contradiction: The mediating role of mixed emotions

PONE-D-21-07959R2

Dear Dr. Velasco,

We’re pleased to inform you that your manuscript has been judged scientifically suitable for publication and will be formally accepted for publication once it meets all outstanding technical requirements.

Kind regards,

Gaëtan Merlhiot

Academic Editor

PLOS ONE
---

## [Editor Report · Acceptance letter]

17 Sep 2021

PONE-D-21-07959R2 

Individual differences in thinking style and dealing with contradiction: The mediating role of mixed emotions 

Dear Dr. Santos:

I'm pleased to inform you that your manuscript has been deemed suitable for publication in PLOS ONE. Congratulations! Your manuscript is now with our production department. 

Kind regards, 

on behalf of

Dr. Gaëtan Merlhiot 

Academic Editor

PLOS ONE